# The transcriptional landscape and diagnostic potential of long non-coding RNAs in esophageal squamous cell carcinoma

Meng Zhou [1,5], Siqi Bao[1,5], Tongyang Gong [2,5], Qiang Wang[3,5], Jie Sun [1], Jiaqi Li[1], Minyi Lu[2], Wanyuan Sun[2], Jianzhong Su [1] ✉, Hongyan Chen [2,4] ✉ & Zhihua Liu [2] ✉

Esophageal squamous cell carcinoma (ESCC) is a deadly cancer with no clinically relevant biomarkers for early detection. Here, we comprehensively characterized the transcriptional landscape of long non-coding RNAs (lncRNAs) in paired tumor and normal tissue specimens from 93 ESCC patients, and identified six key malignancy-specific lncRNAs that were integrated into a Multi-LncRNA Malignancy Risk Probability model (MLMRPscore). The MLMRPscore performed robustly in distinguishing ESCC from normal controls in multiple in-house and external multicenter validation cohorts, including early-stage I/II cancer. In addition, five candidate lncRNAs were confirmed to have non-invasive diagnostic potential in our institute plasma cohort, showing superior or comparable diagnostic accuracy to current clinical serological markers. Overall, this study highlights the profound and robust dysregulation of lncRNAs in ESCC and demonstrates the potential of lncRNAs as non-invasive biomarkers for the early detection of ESCC.

Esophageal cancer (EC) is a digestive system malignancy that seriously threatens human health. The incidence and mortality of esophageal cancer rank 7th and 6th worldwide, respectively[1]. China is a high-risk country for EC, with esophageal squamous cell carcinoma (ESCC) as the predominant subtype, and the incidence and mortality rank 6th and 5th among all malignant tumors, respectively[2]. ESCC is often asymptomatic in the early stages, leading to late-stage diagnosis and poor 5-year survival rates. Endoscopic screening can help early detection, diagnosis, and treatment of ESCC. Epidemiologists from National Cancer Center have undertaken several population-based, multicenter cohort studies in high-risk areas with upper gastro-intestinal cancer in China. The evidence from large population studies has confirmed that endoscopic screening and early intervention is an

effective method to reduce the incidence and mortality of ESCC[3,4]. Moreover, population-based studies have also shown an improvement in the overall 5-year survival rate for ESCC[5]. Nevertheless, endoscopy screening is only being conducted in high-risk areas in China due to a larger cancer burden, lack of hospital personnel, and availability of technology. Therefore, it is imperative to develop reliable biomarkers for the early detection and screening of ESCC.

Long non-coding RNAs (lncRNAs), larger than 200 nucleotides, are recognized as critical mediators of transcriptional regulation, chromatin reorganization, and post-transcriptional regulation. The number of lncRNAs is more abundant than protein-coding RNAs, and their dysregulated expression has been observed in different cancers or even in different subtypes of same cancer, which provides a larger

---

[1]School of Biomedical Engineering, Eye Hospital, Wenzhou Medical University, 325027 Wenzhou, P. R. China. [2]State Key Laboratory of Molecular Oncology, National Cancer Center/National Clinical Research Center for Cancer/Cancer Hospital, Chinese Academy of Medical Sciences and Peking Union Medical College, 100021 Beijing, P. R. China. [3]Department of Anesthesiology, National Cancer Center/National Clinical Research Center for Cancer/Cancer Hospital, Chinese Academy of Medical Sciences and Peking Union Medical College, 100021 Beijing, P. R. China. [4]Key Laboratory of Cancer and Microbiome, National Cancer Center/National Clinical Research Center for Cancer/Cancer Hospital, Chinese Academy of Medical Sciences and Peking Union Medical College, 100021 Beijing, P. R. China. [5]These authors contributed equally: Meng Zhou, Siqi Bao, Tongyang Gong, Qiang Wang. ✉e-mail: sujz@wmu.edu.cn; chenhongyan@cicams.ac.cn; liuzh@cicams.ac.cn

window for finding specific tumor biomarkers[6–9]. Intriguingly, lncRNAs can be extracted from body fluids and are emerging as attractive candidates for the non-invasive "liquid biopsy" approach[10]. Wang et al. have reported that serum levels of HOTAIR could differentiate patients from healthy control with a diagnostic power of 0.793, but the sample size is small[11]. Until now, no systematic and comprehensive study on the lncRNA diagnostic biomarkers from tissue to liquid biopsy has been carried out for ESCC.

In this study, we constructed a diagnostic lncRNA signature based on whole transcriptome data in paired tumor and adjacent normal tissues from patients with ESCC and validated the efficacy of the lncRNA signature in multicenter cross-platform cohorts. Additionally, we have developed cfRNA liquid biopsy diagnostic biomarkers and found that five circulating lncRNAs exhibited superior or comparable diagnostic accuracy in identifying patients with ESCC or esophageal intraepithelial neoplasia (EIN) compared to conventional serological markers.

## Results

### Study design and patient cohorts
We conducted a multicenter, cross-platform clinical discovery and validation study for lncRNA biomarkers in ESCC diagnosis, as shown in Fig. 1. In the discovery phase, we performed a genome-wide screen to identify candidate lncRNA biomarkers in a retrospective case-control cohort of 93 ESCC subjects from Shanxi Cancer Hospital (referred to as the SCH discovery cohort). In the discovery phase, a generalizable *M*alignancy *R*isk *P*robability model (MLMRPscore) was developed by integrating *M*ultiple *L*ncRNA biomarkers to diagnose ESCC in the SCH discovery cohort. In the validation phase, the diagnostic performance of the MLMRPscore was evaluated and examined in different multicenter and cross-platform retrospective cohorts, including three in-house cohorts (referred to as SCH validation cohort, $n = 62$, CAMS tissue cohort, $n = 15$ and CAMS plasma cohort, $n = 77$), two public RNA-seq cohorts from You's study (referred to as You cohort, $n = 23$)[12] and TCGA and GTEx databases (referred to as TCGA-GTEx cohort, 81 ESCC vs. 271 healthy donors), two public microarray cohorts from Li's study (referred to as Li cohort-1, $n = 119$ and Li cohort-2, $n = 60$)[13].

### Identification of ESCC-associated lncRNA biomarkers
To identify ESCC-associated lncRNAs, we analyzed genome-wide lncRNA expression profiling of paired tumor and adjacent normal tissues (ANT) from 93 ESCC patients in the SCH discovery cohort and identified 2103 differentially expressed lncRNAs (DElncRNAs), including 1070 up-regulated and 1033 down-regulated DElncRNAs in tumors compared to normal tissues (Fig. 2A and Supplementary Data 1). As shown in Fig. 2B, the expression pattern of these DElncRNAs was able to distinguish ESCC tissues from non-cancerous tissues (Fig. 2B). To shrink the number of variables and identify the most informative biomarkers, we used RF-RFE algorithms with 10-fold cross-validation and five re-sampling and identified seven DElncRNAs as potential biomarkers. To ensure the reliability and reproducibility of these seven potential lncRNA biomarkers in ESCC, we validated their expression pattern in an external Li cohort-1 (119 ESCC and 119 adjacent non-cancer controls) with a microarray platform and confirmed the same expression variation tendency of six lncRNA biomarkers (AP003548.1, PGM5-AS1, ADAMTS9-AS1, MIR503HG, LINC01082 and LINC03016) in ESCC as revealed in the SCH discovery cohort. MIR503HG is over-expressed, and the remaining five lncRNAs are downregulated in ESCC relative to adjacent non-cancer tissues (Fig. 2C and Supplementary Data 2). Functional enrichment analysis showed that mRNAs co-expressed with six lncRNA biomarkers were enriched in many known cancer-related pathways such as the cGMP–PKG signaling pathway, Apelin signaling pathway, Focal adhesion, and transcriptional misregulation in cancer (Fig. 2D). These results demonstrated their

biological relevance in ESCC pathogenesis and highlighted their potential as a promising diagnostic assay for the detection of ESCC.

### Establishment and verification of a multi-lncRNA diagnostic signature (MLMRPscore) for ESCC in multicenter in-house cohorts
To build a clinically generalizable lncRNA-based malignancy probability model for estimating the risk probability of developing ESCC, we integrated six lncRNA biomarkers to form a multi-lncRNA diagnostic signature (MLMRPscore) that will allow clinicians to assess the risk probability of ESCC using the transformed logistic regression model in the SCH discovery cohort. When tested in another in-house SCH validation cohort of 62 subjects, the MLMRPscore exhibited superior discriminative performance in distinguishing ESCC from non-cancerous tissue with an AUC of 1.000 (Fig. 3A–C and Supplementary Fig. S1).

We next used RT-qPCR assays to measure the expression levels of six lncRNAs in 15 paired ESCC and adjacent non-cancer tissues from the CAMS cohort for verifying the performance of the MLMRPscore (Supplementary Data 3). Consistent with their expression pattern measured by RNA-seq in the SCH discovery and SCH validation cohorts, five lncRNAs (AP003548.1, PGM5-AS1, ADAMTS9-AS1, LINC01082, and LINC03016) were significantly down-regulated and one lncRNA (MIR503HG) were significantly up-regulated in ESCC tissue specimens measured by RT-qPCR assays (Fig. 3E and Supplementary Fig. S2B). The MLMRPscore achieved an AUC of 0.978 (95% CI: 0.931–1.000) with a sensitivity of 93.33% and a specificity of 93.33% (Fig. 3D, F, and Supplementary Fig. S2A). These results initially confirmed the superior and robust performance of the MLMRPscore for its diagnostic potential in tissue specimens.

### Independent validation of the MLMRPscore in external multicenter and cross-platform cohorts
To independently validate the MLMRPscore, we examined the diagnostic performance of the MLMRPscore in four completely blinded external cohorts with different platforms. We analyzed its discriminatory power using a completely blinded external cohort of paired cancerous and non-cancerous tissues of 23 Korean ESCC patients from the You cohort. Results with the You cohort indicated that the MLMRPscore performed robustly in distinguishing ESCC from matched normal controls with an AUC of 0.968 (95% CI: 0.914–1.000) (Fig. 4A–C and Supplementary Fig. S3A). The diagnostic performance of the MLMRPscore was further tested in a large combined cohort (TCGA-GTEx cohort) consisting of 81 ESCC cases and 271 normal esophageal mucosal epithelium tissues. The MLMRPscore identified 70 of 81 ESCC cases and 228 of 271 normal controls with an AUC of 0.951 (95% CI: 0.923–0.978), a sensitivity of 86.42%, and a specificity of 84.13% (Fig. 4D–F and Supplementary Fig. S3C). The expression pattern of these six lncRNAs biomarkers in You and TCGA-GTEx cohorts is consistent, as observed in our different in-house cohorts (Fig. 4B, E and Supplementary Fig. S3B and 3D).

Further validation of the predictive power of the MLMRPscore was subsequently conducted using two independent retrospective case-control cohorts of 119 and 60 ESCC patients from China, respectively. The MLMRPscore was again shown to be capable of distinguishing ESCC from controls, which exhibited an AUC of 0.997 (95% CI: 0.994–1.000, sensitivity: 89.08%, specificity: 99.16%) in the Li cohort-1 (Fig. 4G–I and Supplementary Fig. S4A), and 1.000 (95% CI: 1.000–1.000, sensitivity: 90.00%, specificity: 100.00%) in the Li cohort-2 (Fig. 4J–L and Supplementary Fig. S4C), respectively. In line with the SCH discovery and other validation cohorts, six lncRNA biomarkers revealed consistent dysregulated expression patterns (Fig. 4H and K, Supplementary Fig. S4B and S4D). This multicenter and cross-platform validation study again underscored the reliable and robust diagnostic efficacy of the MLMRPscore.

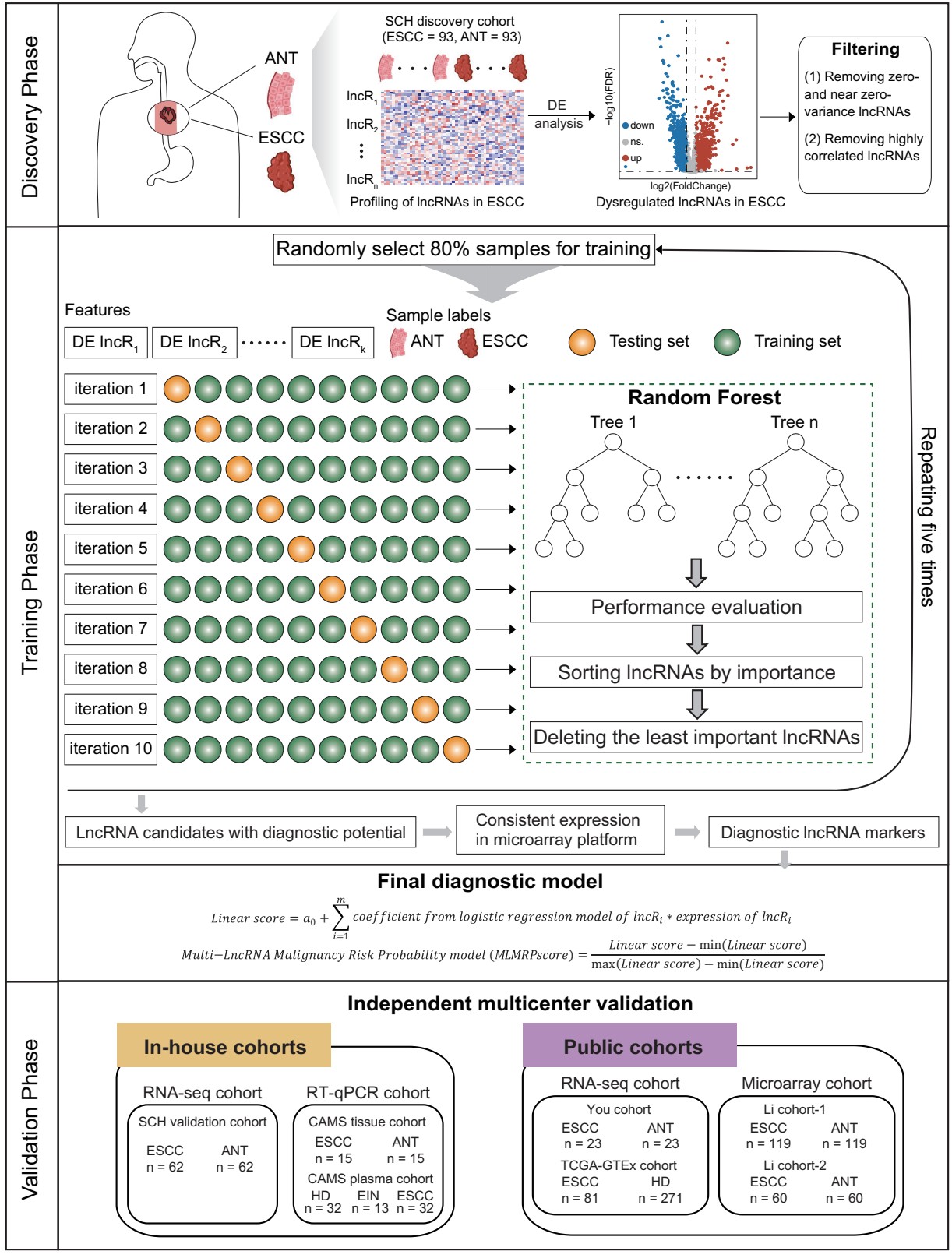

**Fig. 1 | Flow diagram demonstrating the study design.** A multicenter, cross-platform clinical discovery and validation study was conducted for lncRNA biomarkers in ESCC diagnosis. ANT adjacent normal tissues, DE differential expression, ESCC esophageal squamous cell carcinoma, HD healthy donors, lncRNAs long non-coding RNAs.

## The MLMRPscore robustly identifies early-stage I and II ESCC patients

We next examined the relationship between the MLMRPscore and relevant clinical features. The MLMRPscore predicted risk probabilities were significantly higher in patients than in controls, but there were no significant differences between the two groups concerning alcohol use, smoking and gender differences in different cohorts (Supplementary Fig. S5). Tumor diagnosis at earlier stages is

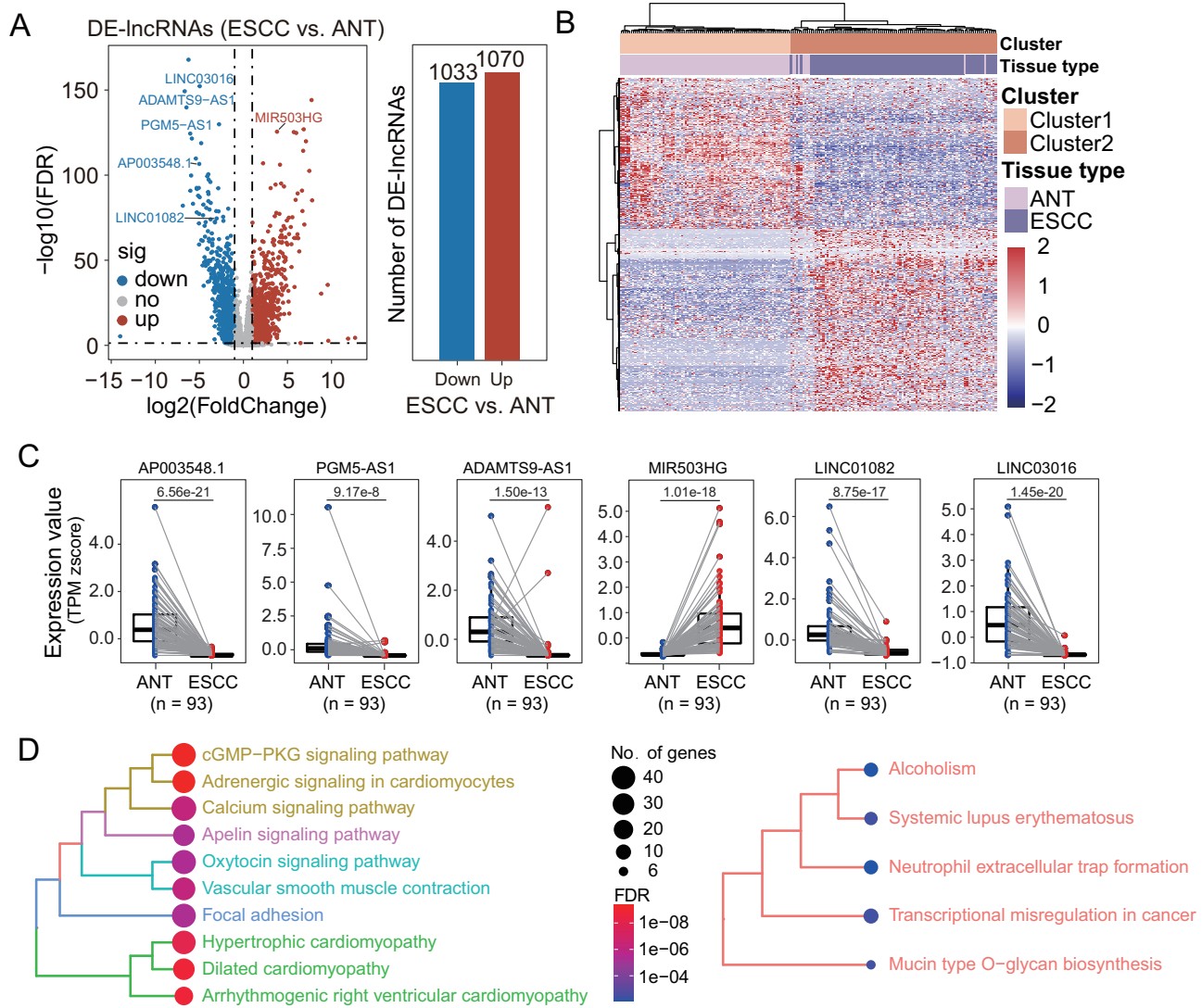

**Fig. 2 | Genome-wide discovery of ESCC-associated lncRNA biomarkers in the SCH discovery cohort. A** Volcano plot showing the log2(fold change) of significantly differentially expressed lncRNAs (FDR-adjusted *p*-value of two-sided Wald test < 0.05 and absolute log2-transformed fold-change > 1) between ESCC tissues from non-cancerous tissues. **B** Heatmap of unsupervised hierarchical clustering of significantly differentially expressed lncRNAs. **C** Boxplots showing expression levels of six lncRNA biomarkers. For the boxplot, the center line indicates the median; box limits indicate the first and third quartiles; whiskers encompass the 1.5× interquartile range. *P* values were determined by two-tailed paired *t*-tests without adjustments for multiple comparisons. **D** KEGG pathway and

GO terms enrichment analysis for mRNAs co-expressed with six lncRNA biomarkers. Jaccard's similarity index was used to measure the pairwise similarities of the enriched GO terms and KEGG pathways. GO terms and pathways with high Jaccard's similarity index are considered similar and clustered into five subsets using the ward.D method. The size of the nodes represents the number of analyzed genes contained in the GO terms or KEGG pathways, and the color represents the FDR-adjusted *p*-value of the enrichment. *P*-values were determined by FDR-adjusted *p*-values of a one-sided version of Fisher's exact test. ESCC, esophageal squamous cell carcinoma; lncRNAs long non-coding RNAs, FDR false discovery rate, ANT adjacent normal tissues.

critical for reducing mortality and improving the prognosis of ESCC. Therefore, we dichotomized patients into stages I and II (early-stage) vs. stages III and IV (late-stage) and evaluated the early diagnostic performance of the MLMRPscore. As shown in Fig. 5, the MLMRPscore also showed superior diagnostic performance in discriminating early-stage I and II ESCC cases from normal controls with AUC of 1.000 (95% CI: 1.000–1.000, sensitivity: 100%, specificity: 100%) in the SCH cohort (Fig. 5A), 0.973 (95% CI: 0.912–1.000, sensitivity: 100.00%, specificity: 93.33%) in the CAMS tissue cohort (Fig. 5B), 1.000 (95% CI: 1.000–1.000, sensitivity: 100.00%, specificity: 95.65%) in the You cohort (Fig. 5C), 0.944 (95% CI: 0.909–0.980, sensitivity: 83.33%, specificity: 92.62%) in the TCGA-GTEx cohort (Fig. 5D), 0.999 (95% CI: 0.997–1.000, sensitivity: 84.91%, specificity: 100.00%) in the Li cohort-1 (Fig. 5E) and 1.000 (95% CI: 1.000–1.000, sensitivity: 85.29%, specificity: 100.00%) in the Li cohort-2 (Fig. 5F)

(Supplementary Fig. S6). These findings collectively demonstrated the potential for the MLMRPscore as a promising early diagnostic tool.

### The non-invasive potential of lncRNA biomarkers in a plasma cohort

To explore the non-invasive potential of the tissue-based lncRNA biomarkers, we measured expression levels of six lncRNA biomarkers in plasma samples of 32 ESCC patients, 32 healthy controls, and 13 EIN patients from the CAMS plasma cohort (Supplementary Data 4). Among the six lncRNAs, five lncRNAs biomarkers (*APO03548.1*, *PGM5-AS1*, *ADAMTS9-AS1*, *LINC01082*, and *LINC03016*) revealed consistent dysregulated expression patterns, as observed in tissue-based cohorts (Fig. 6A). These five lncRNA biomarkers demonstrated robust performance in differentiating ESCC patients from healthy controls, with

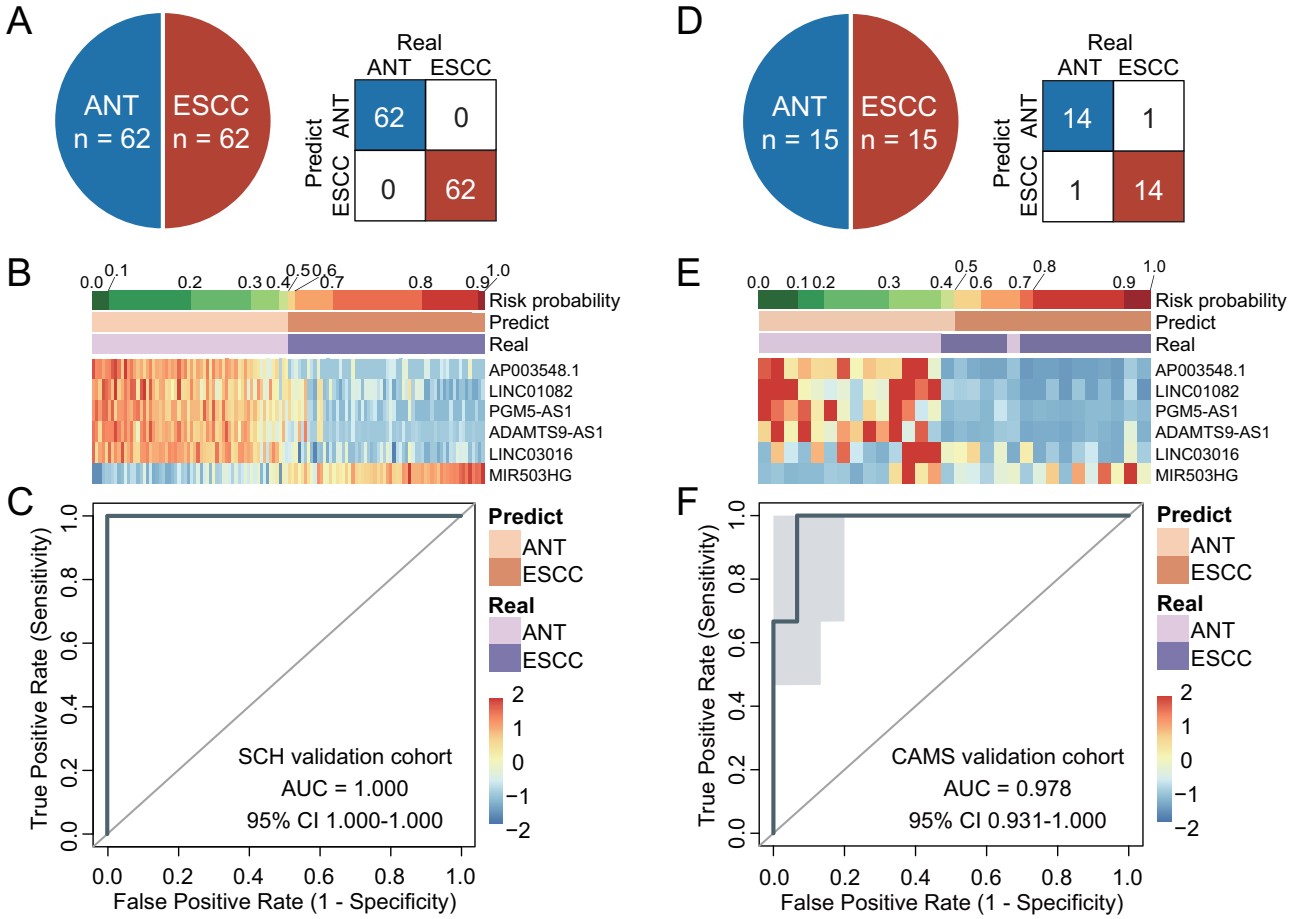

**Fig. 3 | Development and verification of a multi-lncRNA signature (MLMRPscore) for ESCC diagnosis in two independent multicenter in-house cohorts.** A summary of the samples used to validate the performance of MLMRPscore and the confusion matrix in the SCH validation cohort (**A**) and CAMS tissue cohort (**D**). Heatmap of the expression pattern of six lncRNA biomarkers with the corresponding risk probability, predicted label, and true label in the SCH validation cohort (**B**) and CAMS cohort (**E**). Receiver operating characteristic (ROC) curve for the diagnostic performance of the MLMRPscore in the SCH validation cohort (**C**) and CAMS cohort (**F**). In **C**, **F**, data are presented as AUC ± 95% CI. ESCC esophageal squamous cell carcinoma, ANT adjacent normal tissues, AUC area under the curve, CI confidence interval.

AUC values ranging from 0.733 to 0.836, and distinguishing patients with intraepithelial neoplasia from healthy controls, with AUC values ranging from 0.697 to 0.870 (Fig. 6B, C). Furthermore, we compared the diagnostic efficiency of these five lncRNA biomarkers with conventional tumor markers (SCC-Ag, CEA, and CYFRA21-1), and found that the lncRNA biomarkers exhibited superior or comparable diagnostic accuracy in identifying patients with ESCC or EIN compared to the traditional tumor markers (Fig. 6B, C). These results indicate that the lncRNA biomarkers may have the potential as non-invasive tools for early detection of ESCC.

### The use of lncRNA biomarkers provides substantial benefits over current screening approaches

The screening and diagnosis of patients with ESCC have traditionally relied on endoscopic screening or invasive biopsy followed by surgery. To evaluate the clinical benefit of lncRNA biomarkers, we conducted DCA to determine if incorporating these lncRNA biomarkers into clinical decision-making would provide more benefits than harm. As shown in Fig. 7, the DCA curve demonstrated that lncRNA biomarkers achieved higher net benefits than either screening and diagnosing all ESCC patients or none of the patients, across a range of threshold probabilities (Fig. 7). These results indicate that lncRNA biomarkers have the potential to offer greater clinical benefits than either intervening for all cases or not intervening at all, by minimizing the risk of physical harm and misdiagnosis.

## Discussion

LncRNAs, critical regulatory molecules in cancer, have unique advantages in screening tumor diagnostic and prognostic markers due to their wide range of expression panels, high tumor-specificity, and stability in circulating body fluids[6]. Increasing evidence from blood-based studies highlighted potential clinical applications of circulating lncRNAs as passive biomedical tools for early cancer diagnosis[10]. In this study, we conducted a retrospective study and established a six-lncRNA diagnostic signature (MLMRPscore) for the early detection of ESCC.

Among this six-lncRNA signature, *PGM5-AS1* was frequently downregulated in ESCC tissues and exerted a tumor-suppressive function. *PGM5-AS1* was identified as a diagnostic and prognostic biomarker for ESCC patients[14]. *ADAMTS9-AS1* has been identified as a member of lncRNA-signatures for predicting ESCC prognosis and therapeutic response[15,16]. Functional enrichment analysis in our data showed that mRNAs co-expressed with the six lncRNA biomarkers were enriched in many known cancer-related pathways such as cGMP–PKG signaling pathway, Apelin signaling pathway, Focal adhesion and Transcriptional mis-regulation[17–21]. These results further support that candidate lncRNAs were involved in the important biological processes strongly linked to cancer, suggesting that the six lncRNAs have a high potential to construct a diagnostic signature.

The performance of MLMRPscore was assessed by within-cohort cross-validation and external cross-platform validation. MLMRPscore

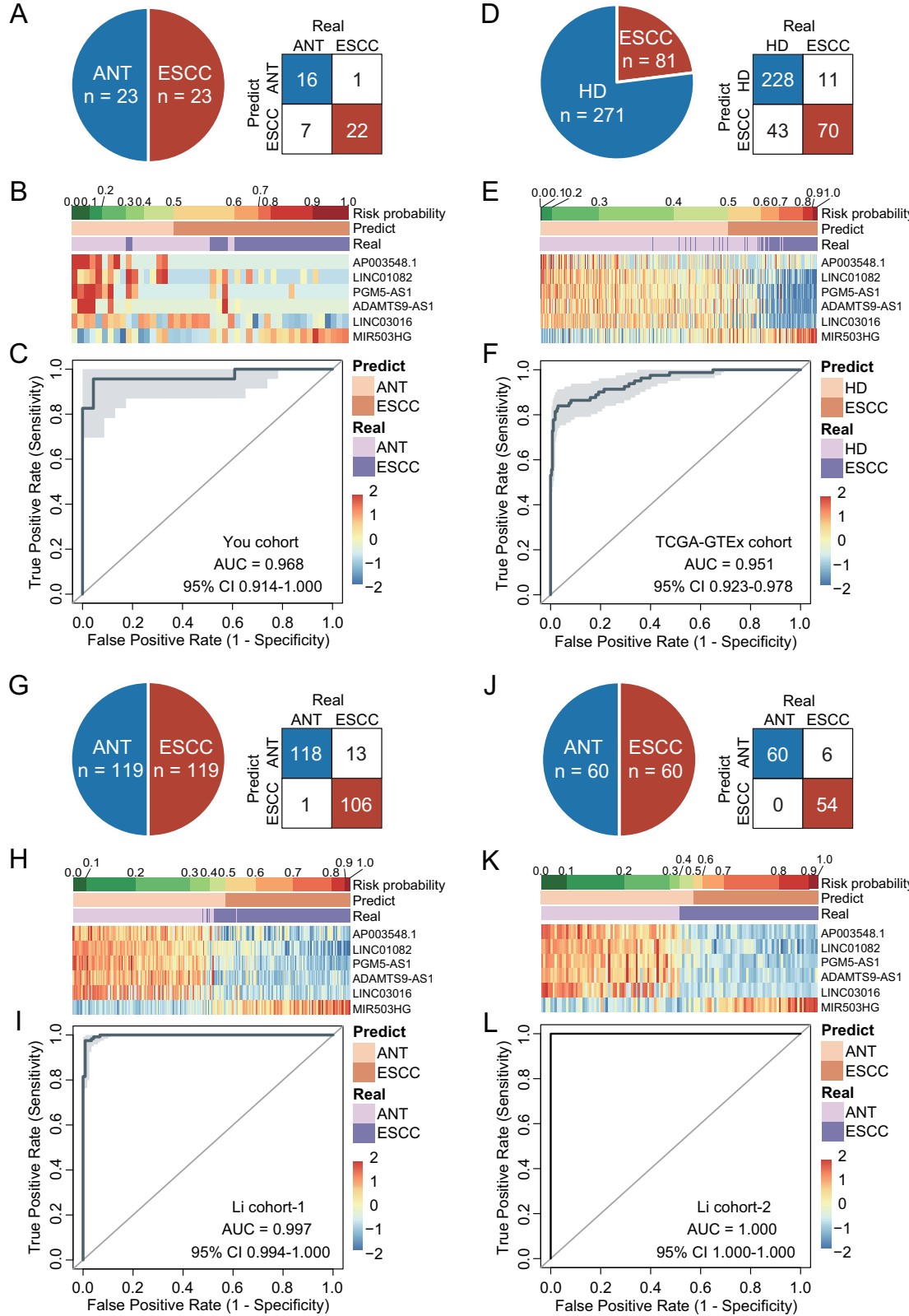

**Fig. 4 | Independent validation of the MLMRPscore in external multicenter and cross-platform cohorts.** A summary of the samples used to validate the performance of MLMRPscore and the confusion matrix in the You cohort (**A**), TCGA-GTEx cohort (**D**), Li cohort-1 (**G**), and Li cohort-2 (**J**). Heatmap of the expression pattern of six lncRNA biomarkers with the corresponding risk probability, predicted label, and true label in the You cohort (**B**), TCGA-GTEx cohort (**E**), Li cohort-1 (**H**), and Li cohort-2 (**K**). Receiver operating characteristic (ROC) curve for the diagnostic performance of the MLMRPscore in the You cohort (**C**), TCGA-GTEx cohort (**F**), Li cohort-1 (**I**), and Li cohort-2 (**L**). In **C**, **F**, **I**, **L**, data are presented as AUC ± 95% CI. ESCC esophageal squamous cell carcinoma, ANT adjacent normal tissues, AUC area under the curve, CI confidence interval, HD healthy donors.

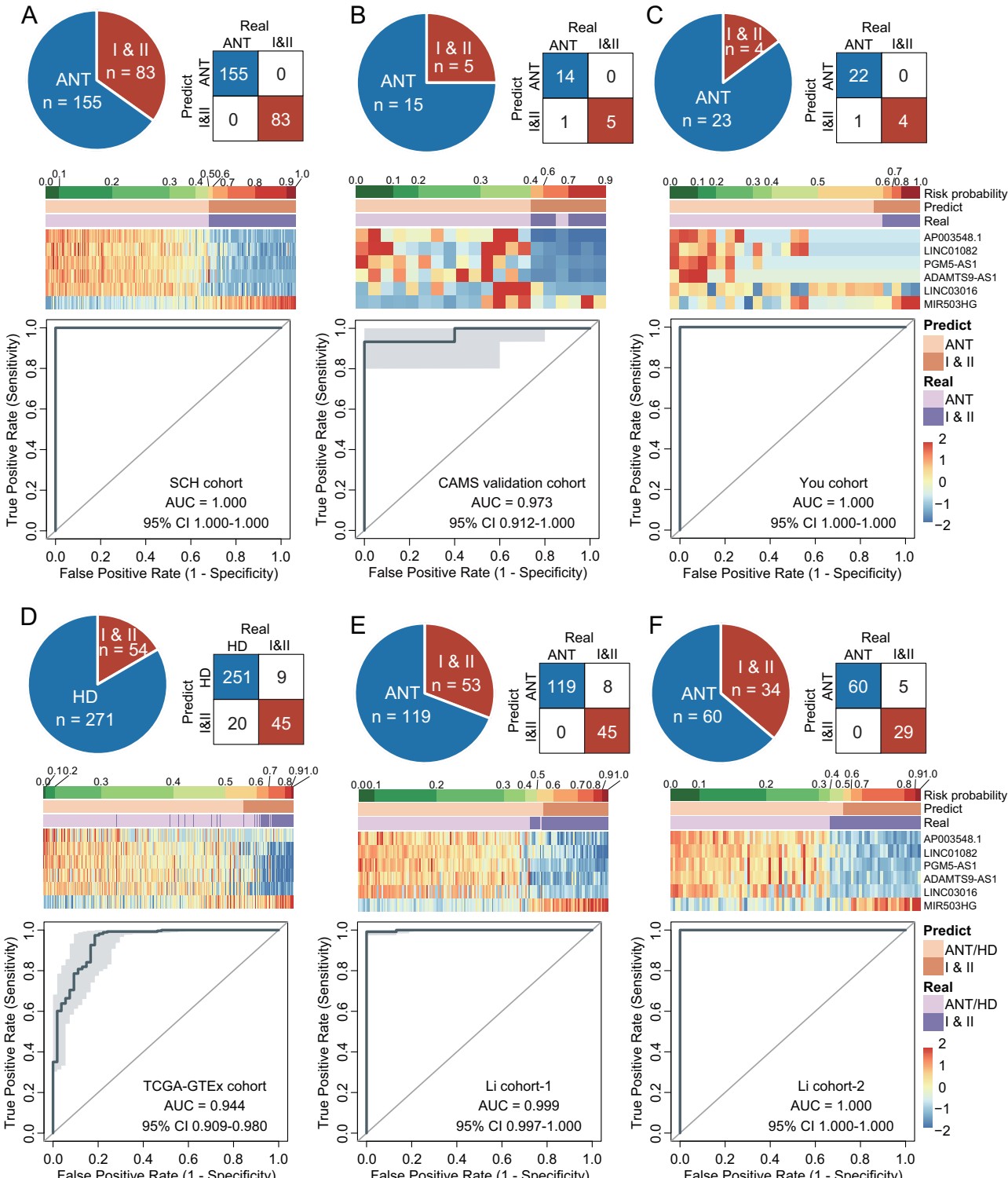

**Fig. 5 | Verification of the early diagnostic performance of the MLMRPscore.**
The confusion matrix, expression heatmap and receiver operating characteristic
(ROC) curve of the MLMRPscore in the SCH cohort (**A**), CAMS tissue cohort (**B**), You
cohort (**C**), TCGA-GTEx cohort (**D**), Li cohort-1 (**E**), and Li cohort-2 (**F**). At the
bottom of **A**–**F**, data are presented as AUC ± 95% CI. ESCC esophageal squamous
cell carcinoma, ANT adjacent normal tissues, AUC area under the curve, CI con-
fidence interval, HD healthy donors.

exhibited robust diagnostic efficacy in different populations and dif-
ferent detection platforms. Notably, the diagnostic performance of the
MLMRPscore was validated in the TCGA-GTEx cohort and three inde-
pendent retrospective case-control cohorts, suggesting MLMRPscore
can effectively discriminate ESCC patients from healthy controls, fur-
ther verifying the robustness of MLMRPscore for diagnostic

biomarkers. Early diagnosis is an effective strategy to improve the
survival and prognosis of ESCC patients[3]. We selected stage I&II
patients from the SCH discovery cohort as an early stage to adjust the
cutoff value of MLMRPscore and then tested it in validation cohorts. As
expected, MLMRPscore showed superior efficacy in the early detection
of ESCC.

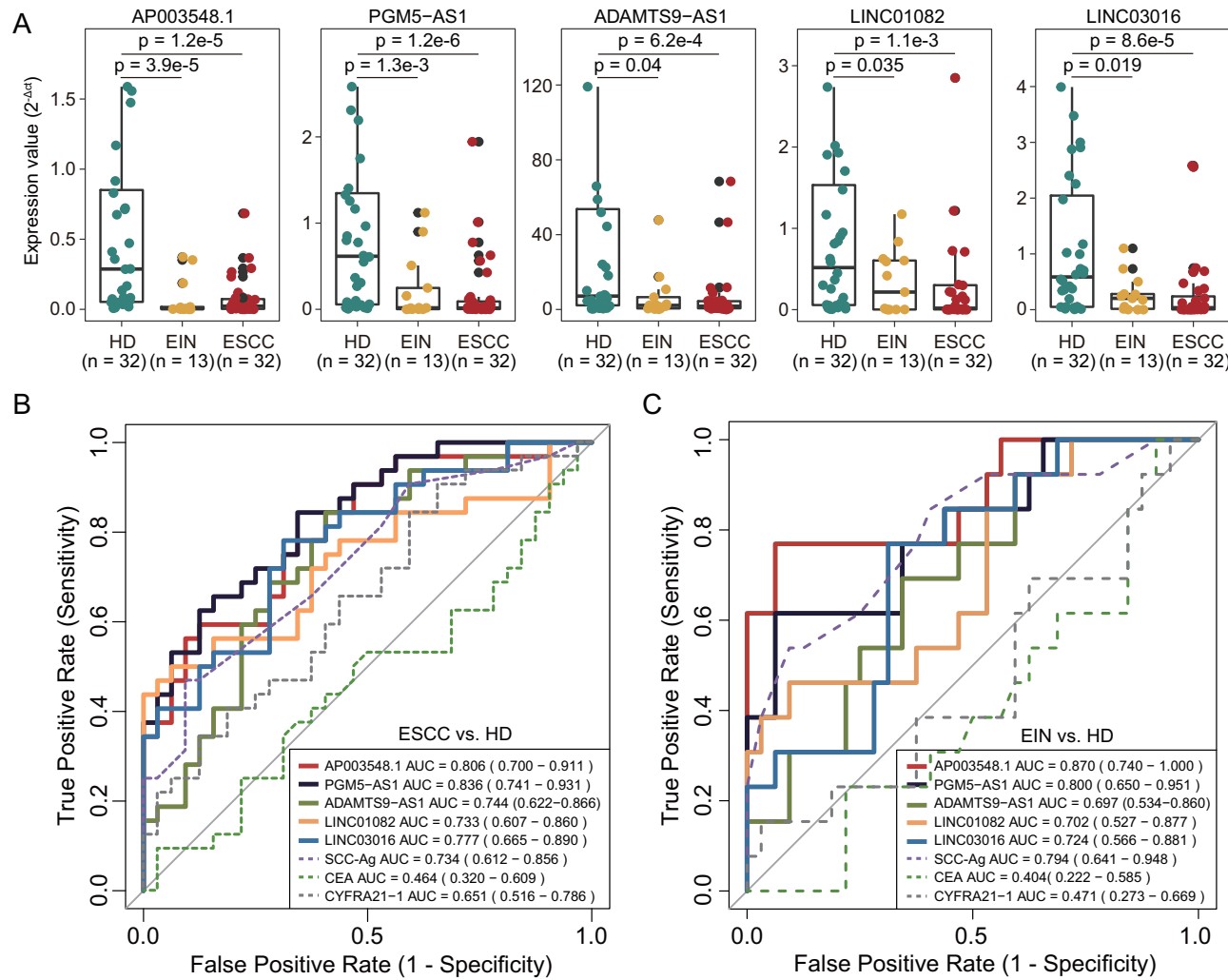

**Fig. 6 | Non-invasive performance of lncRNA biomarkers in a plasma cohort.**
**A** Boxplot showing the expression levels of five lncRNA biomarkers measured in plasma samples of 32 ESCC patients, 32 healthy controls, and 13 patients with intraepithelial neoplasia. ROC analysis for five lncRNA biomarkers and three conventional tumor markers in identifying patients with ESCC (**B**) and intra-epithelial neoplasia (**C**). ESCC esophageal squamous cell carcinoma, EIN esophageal intraepithelial neoplasia, HD healthy donors, AUC area under the curve, CI confidence interval.

Clinical diagnosis requires a simple operation, low cost, and accurate data. Many lncRNAs have been identified in the blood of cancer patients, which could serve as a potential non-invasive diagnostic tool[10,22–24]. Notably, the diagnostic power of circulating lncRNAs has been revealed to be more reliable and superior to conventional glycoprotein markers, circulating tumor cells (CTCs), and cell-free DNA (cfDNA)[10,25]. To explore the non-invasive potential of the MLMRPscore, we assessed the expression of lncRNAs among the MLMRPscore model in our institute plasma cohort using RT-qPCR. Intriguingly, five circulating lncRNAs exhibited consistent dysregulated expression patterns with tissue lncRNAs. These five lncRNAs demonstrated robust performance in differentiating patients with ESCC and EIN from healthy controls. In addition, these five lncRNAs exhibited superior or comparable diagnostic accuracy compared to conventional clinical serological biomarkers, including SCC-Ag. These results suggest that five circulating lncRNAs may have the potential for early detection of ESCC.

Liquid biopsy-based early detection technology provides promising opportunities for early detection of ESCC. However, the translation of liquid biopsy into clinical practice for early cancer detection will be challenging. No testing of liquid biopsy-based screening is currently being used in clinical practice. There are

several limitations to this study. Due to data constraints, we only tested the diagnostic performance of five circulating lncRNAs on our recently collected plasma samples from a limited number of patients and healthy controls. Several steps need to be taken to apply our approach as an important supplement to the current cancer screening methods or as a screening tool. Firstly, the diagnostic efficacy of circulating lncRNAs should be verified in long-time, large-scale, multicenter, retrospective, and prospective plasma cohorts. Importantly, carefully controlled trials, and potential comparison with and integration into conventional endoscopic screening in the intended use population needed to be performed. In addition, given that the main aim of our present study was to identify diagnostic biomarkers for ESCC, we are unable to evaluate whether these markers could also monitor tumor progression or predict response to treatment in ESCC patients. We will pursue such studies in the future.

In conclusion, our study utilized a systematic, multicenter, cross-platform clinical biomarker discovery and validation framework to develop a stable and powerful multi-lncRNA diagnostic signature (MLMRPscore) capable of accurately identifying patients with ESCC, including early-stage tumors in clinical cohorts. This diagnostic signature was successfully validated in different independent tissue cohorts. Five circulating lncRNAs based on the MLMRPscore showed

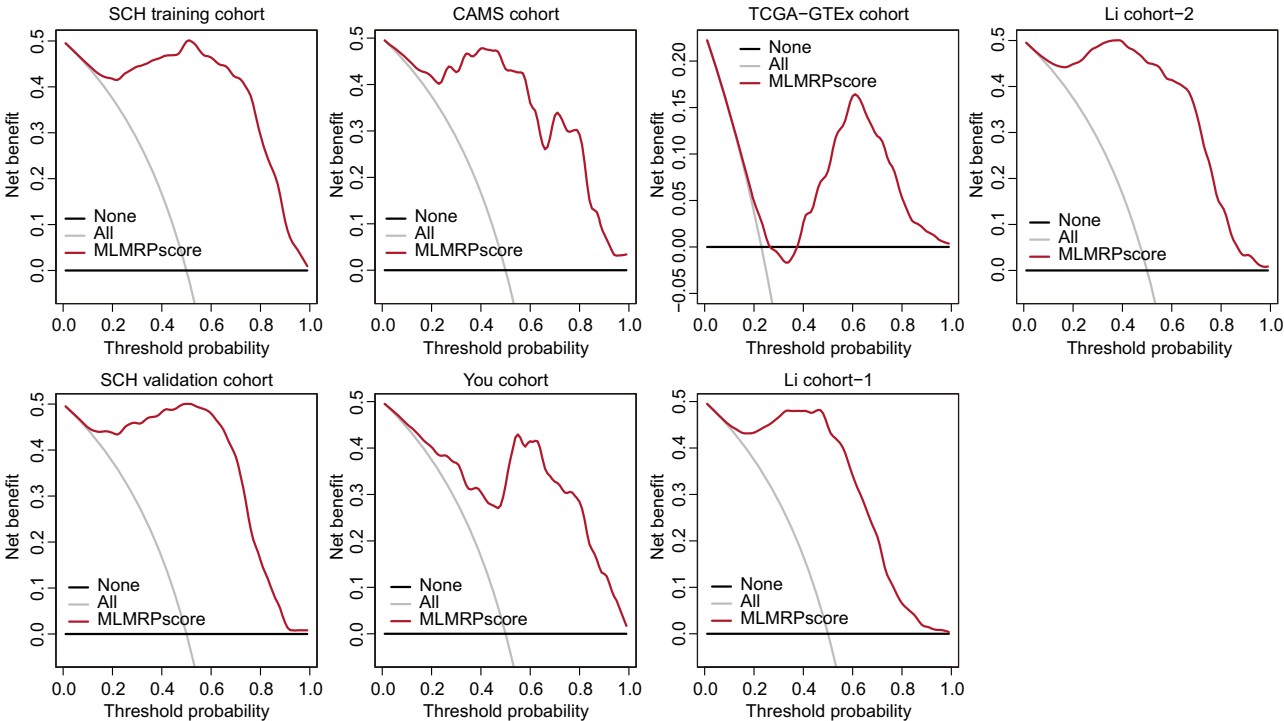

**Fig. 7 | LncRNA biomarkers provide substantial benefits over current screening approaches.** Decision curve analysis to evaluate the clinical benefits of lncRNA biomarkers in multiple cohorts.

robust performance in identifying patients with ESCC and EIN from healthy controls in a plasma cohort, laying the foundations for future non-invasive ESCC detection methods.

## Methods

### Patient biospecimens collection and preparation

This study was performed according to the Declaration of Helsinki and approved by the Ethics Committee of the Cancer Hospital of the Chinese Academy of Medical Sciences (CAMS) and the Shanxi Medical University and Shanxi Cancer Hospital. Informed consent was obtained from all subjects, and all data were anonymously analyzed.

155 ESCC patients at the Shanxi Cancer Hospital (SCH), 32 ESCC patients, 13 EIN patients, and 32 healthy subjects at the CAMS were recruited in this study. All the patients underwent oesophagectomy or endoscopic surgery and received no chemotherapy or radiotherapy before surgery. The paired tumor and adjacent normal tissues from each individual were collected at the time of treatment. All cases were classified according to the WHO histological classification criteria. Hematoxylin–eosin (H&E)-stained sections from each specimen were reviewed by three pathologists to confirm the original diagnosis and stage. Clinical and pathologic data used in this study were identified by a retrospective review of the electronic medical record. Tissue samples (ESCC and adjacent normal tissues) were immediately placed in RNAlater (Qigen, Germany) after esophagectomy, then stored at −80 °C.

### Public multicenter patient cohorts

For the biomarker validation phase of our study, we also collected ESCC data from multicenter cohorts previously published, including paired cancerous and non-cancerous tissues of 23 Korean ESCC patients from You et al. (You cohort)[12], paired cancer and adjacent normal tissues of 119 and 60 Chinese ESCC patients from Li et al. (Li cohort-1 and Li cohort-2)[13], 81 ESCC generated by The Cancer Genome Atlas (TCGA)[26] and 271 normal esophageal mucosal epithelium tissues obtained from GTEx project[27] (TCGA-GTEx cohort).

### High-throughput data processing and lncRNA expression mining

Transcriptome data for patients were obtained from our previous study[28]. RNA-seq reads were aligned to an index of the human genome (hg38) using STAR-2.6.1. Transcript quantification at the gene level was performed using htseq-count against the Ensembl GRCh38 release 95.gtf file. Finally, we generated expression profiles of 16,064 lncRNAs of paired tumor and non-cancerous tissues from 155 ESCC patients at SCH (Supplementary Data 5).

Processed lncRNA expression data profiled by the Agilent human lncRNA+mRNA array V.2.0 platform of patients in Li cohort-1 and Li cohort-2 were obtained from GSE53624 and GSE53622 through the GEO database. The probe sequences of the Agilent human lncRNA+mRNA array V.2.0 platform were also downloaded from the GEO database. Bowtie2 software was used to align probe sequences to the human reference genome (GRCh38/hg38). Bedtools was used to compare the position of the probe and the mRNA and lncRNA in the GRCh38 annotation file downloaded from the Ensembl genome browser to filter the probes located in the region of lncRNA and mRNA. Probes that mapped uniquely to the lncRNA or mRNA transcripts with no mismatches were kept, resulting in the inclusion of 18,021 mRNAs and 6138 lncRNAs. Using the probe-gene annotation file, the probe expression file of GSE53624 and GSE53622 were transformed to the expression level of lncRNA and mRNA.

The RNA-seq count data of patients in the You cohort were downloaded from GSE130078 through the GEO database. The raw count was converted to the transcripts per million (TPM) level for further analysis.

The processed RNA-seq data of patients and normal esophageal mucosal epithelium tissues in the TCGA-GTEx cohort were retrieved from UCSC Xena (http://xena.ucsc.edu/). FPKM (fragments per kilobase of transcript per million fragments mapped) expression levels were converted to transcripts per million (TPM) levels for further analysis.

## Tissue RNA extraction and RT-qPCR analysis

The total RNA of 15 paired tumors and adjacent non-cancer tissues (clinical information listed in Supplementary Data 3) was extracted using the Trizol reagent. Briefly, tissues were frozen rapidly using liquid nitrogen and ground to powder. Total RNA was then released with Trizol, extracted by chloroform, and precipitated by isopropyl alcohol. The precipitate was washed with 75% ethanol and dissolved in RNase-free water. Total RNA was reverse transcribed into cDNA using lnRcute lncRNA First-Strand cDNA Kit (Cat#KR202-2, TIANGEN). Real-Time Quantitative Polymerase Chain Reaction (RT-qPCR) was performed using TaqMan™ Fast Advanced Master Mix (Cat#4444557, Thermo Fisher Scientific). GAPDH was used as an internal reference to calculate $2^{-\Delta Ct}$, representing the relative expression level of lncRNAs. TaqMan probes were purchased from Thermo Fisher Scientific (GAPDH Cat#4326317E, AP003548.1 Cat#4351372, LINC01036 Cat#4426961, LINC01082 Cat#4351372, MIR503HG Cat#4426961, ADAMTS9AS1 Cat#4426961, PGM5AS1 Cat#4426961).

## Plasma collection, RNA extraction, and RT-qPCR

Plasmas were collected in our institute from patients with ESCC or esophageal intraepithelial neoplasia (EIN) (Supplementary Data 4). Plasma RNA was extracted using QIAzol reagent (QIAGEN Cat#79306). Briefly, 1 ml QIAzol reagent was required per 200 µl plasma and mixed by pipetting. Then incubated at room temperature for 5 min. RNA was extracted into the aqueous phase using 200 µl chloroform and precipitated by isopropanol, including 1/10 volume NaCl (3 M, pH 5.3) and 1 µl glycogen at −20°C overnight. Centrifuged to collect RNA precipitate, washed the pellet with 75% ethanol, and dissolved using 17 µl nuclease-free water. 8 µl RNA was reverse transcribed into cDNA and RT-qPCR was performed using PowerUp™ SYBR™ Green Master Mix (Thermo Fisher Scientific). The primers were listed in Supplementary Data 6.

## Statistical analysis

All the statistical analyses in this study were conducted using software R, version 3.5.1 and Bioconductor. The random forest-recursive feature elimination (RF-RFE) algorithms with 10-fold cross-validation and five re-sampling were used to shrink the number of variables and identify the most informative biomarkers. A *M*ulti-*L*ncRNA *M*alignancy *R*isk *P*robability model (MLMRPscore) was developed to estimate the malignancy risk probability of developing ESCC using the transformed logistic regression scoring model. The training distribution matching (TDM) method was used for cross-platform and cross-study normalization using the R package 'TDM' (v0.3)[29]. The receiver operating characteristic (ROC) curves and area under the curve (AUC) values were used to assess the diagnostic efficiency of the MLMRPscore. The 95% confidence interval (CI) for individual AUCs was computed with 2000 stratified bootstrap replicates using the "ci.sp" function as implemented in the R package 'pROC' (v1.18.0). The sensitivity, specificity, and confusion matrices were also calculated to summarize the discriminatory performance of the MLMRPscore. Decision curve analysis (DCA) was conducted to evaluate the clinical benefit of prediction models. In the discovery phase, differentially expressed lncRNAs between cancer and normal tissue were identified using the R package 'DESeq2' (v1.34.0) with the fold change >2 or <0.5 and false discovery rate (FDR) adjusted *p*-value < 0.05. The two-tailed paired *t*-tests were performed to compare the values of two samples taken from the same individual, and the Mann–Whitney *U* test was performed to compare the values between two groups of samples through the R function "wilcox.test". Unsupervised clustering was conducted using the R package 'pheatmap' (v1.0.12) with ward.D and Canberra as metrics. KEGG pathway enrichment was performed using the R package 'clusterProfiler' (v4.2.2)[30]. All statistical analyses were two-sided, and a *p*-value < 0.05 was considered statistically significant.

## Reporting summary

Further information on research design is available in the Nature Portfolio Reporting Summary linked to this article.

## Data availability

The raw transcriptome data of this study have been deposited in the Genome Sequence Archive of Beijing Institute of Genomics, Chinese Academy of Sciences with accession number HRA003107. According to the guidelines of GSA-human, all non-profit researchers and the Principle Investigators of any research group are allowed access to the data. Request for this data access should be addressed to the corresponding author Zhihua Liu (liuzh@cicams.ac.cn). The remaining data are available within the article, supplementary information, and Source data. LncRNA expression profiles generated during this study are provided in Supplementary Data 5. All public ESCC data are available from the Gene Expression Omnibus (GEO) database under accession number GSE53624, GSE53622 and GSE130078 and UCSC Xena (https://xenabrowser.net/datapages/?cohort=GDC%20TCGA%20Esophageal%20Cancer%20(ESCA)&removeHub=https%3A%2F%2Fxena.treehouse.gi.ucsc.edu%3A443). Source data are provided with this paper.

## Code availability

The source code of this work can be downloaded from GitHub (https://github.com/ZhouSunLab-Workshops/MLMRP).

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

## Acknowledgements

This study was supported by the National Natural Science Foundation of China (82188102 to Z.L.), the National Key Research and Development Program of China (2021YFC2501000 to Z.L., 2021YFA1101200 to Z.L., 2022YFC3401003 to H.C.), the fund of Guangdong Basic and Applied Basic Research Foundation (2019B030302012 to Z.L.), Beijing Natural Science Foundation (7212085 to H.C.), the CAMS Innovation Fund for Medical Sciences (2021-I2M-1-018 to Z.L., 2021-I2M-1-067 to H.C.), the Fundamental Research Funds for the Central Universities (3332021091 to Z.L.) and the Non-profit Central Research Institute Fund of Chinese Academy of Medical Sciences (2019PT310027 to H.C.). The funders had no roles in study design, data collection and analysis, publication decisions, or manuscript preparation.

## Author contributions

Z.L., H.C., J.Z.S., and M.Z. contributed to conception and design; M.Z., S.B., J.S., Jie Sun, and J.L. contributed to data analysis and interpretation. T.G., Q.W., M.L., and W.S. contributed to the provision of study materials or patients and experiments. M.Z., S.B., H.C., and T.G. drafted the manuscript. All authors read and approved the final manuscript.

## Competing interests

The authors declared no competing interests.
