## [Peer Review File · Nature Communications]

REVIEWER COMMENTS

Reviewer #1 (Remarks to the Author): clinical expertise in oesophageal squamous cell carcinoma

The authors report a panel of non-coding mRNAs that may identify esophageal squamous cancer on a blood-based panel. The authors conclusions, however, are somewhat simplistic and vastly overstated. Such a panel is no way ready to apply for population-based screening, even in a high incidence area for this disease. Carefully controlled trials, and potential comparison with and integration into conventional endoscopic screening, need to be performed. Currently there are no established endoscopic screening programs for either esophageal or gastric cancers underway in China. Also, focusing solely on one cancer is not the direction of blood-based population wide screening for early detection of cancers, rather this approach is being broadly applied for pan cancer screening. The authors never make clear what is the realistic potential of such a screening tool, this is a major flaw of this report. Much editorializing about this tool, without data or discussion to support claims, needs to be taken out, including statements about "great potential" and that application is already "warranted in clinical practice." These are gross overstatements without basis in fact or real practice potential. A "large clinical cohort" leading to real clinical practice would likely involve thousands or tens of thousands of cases in the context of a carefully controlled clinical trial.

Specific comments are outlined below:

Abstract: The statement that earlier detection improves outcome in esophageal squamous cancers is a simplistic statement. This is a highly virulent systemic disease which carries a poor prognosis even in early-stage disease. Endoscopic screening programs have not been established and there are no data to support efficacy of screening or early detection of this cancer in high-risk populations. There is no comment throughout this manuscript about endoscopic screening, detection and intervention in precancerous dysplasia, and other issues. The statement that this tool has "great potential" is a vast overstatement. No data is provided how this would be incorporated or practically used in a functional screening program.

Introduction: What would a positive test lead to? Endoscopy? What would be the cost of screening 100s of thousands or millions of patients, for this solitary cancer, as opposed to endoscopic screening? This issue is never addressed. True validation here would be the application in a large-scale clinical trial, which is not even broached.

Patient population: Why did patients only undergo surgery? This is a poor prognosis cancer in which the added role of chemotherapy and radiotherapy is clearly established.

Results: The literature is already full of prognostic markers in this and other cancers, it is unclear what this tool offers beyond conventional use of T and N staging and response to preoperative therapy. The authors make no comment about clinical treatment of patients with this disease.

Discussion: This is largely unfocused and scattershot and lumps in various different cancers, much of this discussion is irrelevant. It remains highly speculative that blood based genomic evaluation for this, or any cancer has any realistic or practical application at the current time. True validation of such testing would be in the context of large-scale screening endeavors in which this biomarker is incorporated into conventional endoscopic screening. Unfortunately, prognostic markers currently have little utility in this disease, and arguably more important are predictive biomarkers of therapeutic intervention. The simplistic statement that the measure here was "independent" of conventional risk factors such as gender, smoking, or alcohol use provides little support for use of this marker. A few hundred patients are hardly a "large" clinical cohort that supports broad application of a likely expensive and really untested biomarker.

Reviewer #2 (Remarks to the Author): expertise in transcriptomics of oesophageal cancer

In this study, Zhou and colleagues represented a good effort to analyze the transcriptional

landscape and diagnostic potential of long non-coding RNAs in ESCC through conducting a multicenter, retrospective study. Novel lncRNA biomarkers and diagnostic model were identified using a machine-learning-based discovery strategy, and validated in multicenter, cross-platform retrospective cohorts, indicating the robustness and soundness of the lncRNA diagnostic model. Overall, this is a well-performed study that focused on a very significant issue. The data are novel, clear and well-presented. Importantly, this study provides an effective tool that has a great potential for the early diagnosis of ESCC. However, there are several minor concerns need to be addressed to further strengthen this study.

1. How many lncRNAs were measured in the discovery dataset? For the multiple comparisons, the authors need to adjust the p-values by considering the comparisons of all the lncRNAs, and then identify the differentially expressed lncRNAs based on the adjusted p-values. Also, how to select six lncRNAs from significantly differentially expressed lncRNAs? Did they have the smallest p-values?
2. The reported predictive power of the six lncRNA signature is impressive. Among them, five lncRNAs were downregulated in ESCC. While aberrant keratinocyte differentiation is considered to be a key mechanism in the initiation of ESCC, which is driven by downregulation of differentiation-associated genes. Whether these lncRNAs influence cell differentiation and development of ESCC? In addition, whether these six lncRNAs are specifically expressed in the esophagus tissues? What about the expression of 6 lncRNAs in other tissues?
3. Is it likely that some of the six lncRNAs correlate with each other? It would be better to compare the diagnostic performance between single lncRNA biomarker and their combination.
4. Interestingly, the authors showed that the expression of 6 lncRNAs can robustly identify early-stage I and II ESCC patients. What's the correlation between the expression of 6 lncRNAs and clinical stages of ESCC? Whether the expression of 6 lncRNAs has the potential to monitor the progression of ESCC?
5. In the process of model verification, the author uses the same threshold. This value is judged according to the ROC curve. Why not recalculate the AUC in the validation data and select the corresponding threshold?
6. While the study is multicenter, whether to apply multilevel approaches to consider levels.
7. The expression of MIR503HG in tumor samples is higher than that in adjacent normal tissues. Whether MIR503HG is expressed in esophageal cancer cells or normal esophageal epithelial cells?
8. What R Bioconductor packages were used?
9. What methodology was used for the division of the training set?
10. Whether these six lncRNAs or at least one of six showed any association with other human diseases?

Reviewer #3 (Remarks to the Author): expertise in lncRNA biomarker signature bioinformatics

General comments

In this study, the authors developed a classifier comprised of six lncRNAs for estimating the risk probability of developing ESCC. These biomarkers were identified in tissue specimens and were demonstrated in six independent, multicenter, cross-platform cohorts. Furthermore, the classifier was unaffected by other relevant clinical features and showed superior efficacy in the early diagnosis of ESCC. In general, the paper was well written. The following are some specific concerns.

Major points

- (1) As the authors mentioned in the manuscript, lncRNAs can be used as noninvasive biomarkers. Have the authors validated cell-free forms of these six lncRNAs in the body fluids (plasma or serum) with RT-qPCR assays? After all, tissue biomarkers are limited by the invasive sampling approach.
- (2) Although the superior performance of the lncRNA panel was well demonstrated, it is better to compare the diagnostic performance of the lncRNA panel with currently established methods (gastroscopy, X-ray imaging, or other biomarkers) for ESCC diagnosis to explain the advantage of

this lncRNA panel. And the authors may describe more clearly the added value of the lncRNA panel to guide ESCC early intervention programs compared to currently established methods.

(3) In the training phase (Fig1), lncRNA candidates with diagnostic potential identified by random forest were filtered by consistent expression in the microarray platform. It seems that the microarray platform referred to the independent validation cohort (Li cohort-1) according to the description of lines 220-222 in the manuscript. If so, partial information of the independent validation cohorts has been leaked to the training process, which will overfit the prediction model.

(4) How the six lncRNAs were selected in 2103 differentially expressed lncRNAs, and why six? The authors may describe more clearly about feature selection.

Minor points

(1) In lines 51 and 314, the statement "personalized" is inappropriate because there is no customized panel for each patient.

(2) Primers used in the CAMS cohort need to be provided.

(3) In Fig2D, the meaning of the color for KEGG pathways and GO terms and the clustering is unclear.

(4) "six lncRNA biomarkers" instead of "six lncRNAs biomarkers" in line 271.

(5) Unnecessary bold font for "curve" in line 604.

(6) "Consistent expression in microarray platform" instead of "Consistent expression in microarray platform" in Fig1.

Response to the Reviewers' Comments

Response to Reviewer #1

Reviewer #1 (Remarks to the Author): clinical expertise in oesophageal squamous cell carcinoma

The authors report a panel of non-coding mRNAs that may identify esophageal squamous cancer on a blood-based panel. The authors conclusions, however, are somewhat simplistic and vastly overstated. Such a panel is no way ready to apply for population-based screening, even in a high incidence area for this disease. Carefully controlled trials, and potential comparison with and integration into conventional endoscopic screening, need to be performed. Currently there are no established endoscopic screening programs for either esophageal or gastric cancers underway in China. Also, focusing solely on one cancer is not the direction of blood-based population wide screening for early detection of cancers, rather this approach is being broadly applied for pan cancer screening. The authors never make clear what is the realistic potential of such a screening tool, this is a major flaw of this report. Much editorializing about this tool, without data or discussion to support claims, needs to be taken out, including statements about “great potential” and that application is already “warranted in clinical practice.” These are gross overstatements without basis in fact or real practice potential. A “large clinical cohort” leading to real clinical practice would likely involve thousands or tens of thousands of cases in the context of a carefully controlled clinical trial.

Response: We do appreciate for your constructive and detailed comments. We acknowledge that the conclusions of our study may have been overstated and that additional studies, including non-invasive potential, need to be conducted before our proposed lncRNA panel can be applied for population-based screening. We complemented the experiments to validate the non-invasive potential of lncRNA biomarkers in a plasma cohort. And we also performed decision curve analysis (DCA) to evaluate the clinical benefit of lncRNA biomarkers.

Regarding the potential of our proposed lncRNA biomarkers, we agree that further studies are needed to determine its realistic potential, and we acknowledge that we did not provide sufficient discussion on this topic in our manuscript. We will revise our manuscript to more accurately describe the potential of our lncRNA biomarkers and the need for further studies to determine its utility in clinical practice.

We also appreciate the reviewers' comments on the need for controlled clinical

trials and the comparison of our screening tool with conventional endoscopic screening. We will take these suggestions into account and include more discussion on these topics in our revised manuscript.

Finally, we apologize for any editorializing in our manuscript that may have overstated the potential of our lncRNA biomarkers without clinical trials to support our claims. We will carefully revise our manuscript to ensure that all claims are supported by the data and accurately reflect the potential of our lncRNA biomarkers.

We have carefully and thoroughly revised the manuscript according to the comments of the reviewer. The revisions are as follows.

- 1) To explore the non-invasive potential of the tissue-based lncRNA biomarkers, we examined the diagnostic performance of the tissue-based lncRNAs panel in a plasma-based biomarker cohort. We measured expression levels of six lncRNA biomarkers in plasma samples of 32 ESCC patients, 32 healthy controls, and 13 EIN patients from the CAMS plasma cohort. Among the six lncRNAs, five lncRNAs biomarkers (AP003548.1, PGM5-AS1, ADAMTS9-AS1, LINC01082 and LINC03016) revealed consistent dysregulated expression patterns, as observed in tissue-based cohorts. These five lncRNA biomarkers demonstrated robust performance in differentiating ESCC patients from healthy controls, with AUC values ranging from 0.733 to 0.836, and in distinguishing patients with intraepithelial neoplasia from healthy controls, with AUC values ranging from 0.697 to 0.870. Furthermore, we compared the diagnostic efficiency of these five lncRNA biomarkers with conventional tumor markers (SCC-Ag, CEA, and CYFRA21-1), and found that the lncRNA biomarkers exhibited superior or comparable diagnostic accuracy in identifying patients with ESCC or intraepithelial neoplasia compared to the traditional tumor markers. These results indicate that the lncRNA biomarkers may have the potential as non-invasive tools for early detection of ESCC. We have added new data in Figure R1A-C.

Figure R1. Non-invasive performance of lncRNA biomarkers in a plasma cohort. (A) Boxplot showing the expression levels of five lncRNA biomarkers measured in plasma samples of 32 ESCC patients, 32 healthy controls, and 13 patients with intraepithelial neoplasia. ROC analysis for five lncRNA biomarkers and three conventional tumor markers in identifying patients with ESCC (B) and intraepithelial neoplasia (C).

2) The screening and diagnosis of patients with ESCC have traditionally relied on endoscopic screening or invasive biopsy followed by surgery. To evaluate the clinical benefit of lncRNA biomarkers, we conducted DCA to determine if incorporating these lncRNAs biomarkers into clinical decision-making would provide more benefits than harm. The DCA curve demonstrated that lncRNA biomarkers achieved higher net benefits than either screening and diagnosing all ESCC patients or none of the patients, across a range of threshold probabilities. These results indicate that lncRNA biomarkers have the potential to offer greater clinical benefits than either intervening for all cases or not intervening at all, by minimizing the risk of physical harm and misdiagnosis. We have added new data in **Figure R2**.

Figure R2. Decision curve analysis to evaluate the clinical benefits of lncRNA biomarkers.

- 3) We reorganized the abstract, results and discussion, corrected the overstated and inappropriate descriptions such as “great potential” “warranted in clinical practice.” and “large clinical cohort” etc.
- 4) For cancer, early detection and treatment are very important. Endoscopic screening can potentially prevent upper gastrointestinal cancers by early diagnosis and early treatment and has been widely adopted in screening programme. In China, epidemiologists have performed a population-based, multicentre cohort study in high risk areas with upper gastrointestinal cancer and shown that one-time endoscopic screening programme was associated with a significant decrease in upper gastrointestinal cancer incidence and mortality ^{1,2}. However, endoscopy screening is only being conducted in high risk areas in China due to a larger cancer burden, lack of the hospital personnel, and availability of technology. Therefore, there is an imperative need to develop reliable biomarkers for early detection and screening of ESCC. Liquid biopsy-based early detection technology provides opportunities for early detection of ESCC. We emphasized the strategies for our identification of more precise and reliable biomarkers for ESCC from tissue to liquid biopsy. We have complemented the descriptions in the Introduction section.
- 5) We corrected the overstatements and reorganized the discussion. We complemented

the discussion on the diagnostic efficacy of circulating lncRNAs in ESCC and EIN. We also described the limitations of this study. Due to data constraints, we only tested the diagnostic performance of five circulating lncRNAs on our recently collected plasma samples, in which the number of patients and healthy controls was limited. Several steps need to be taken in order to apply our approach as an important supplement to the current cancer screening methods or as a screening tool. Firstly, the diagnostic efficacy of circulating lncRNAs should be verified in a large, multicenter, retrospective and prospective plasma cohort. Importantly, carefully controlled trials, and potential comparison with and integration into conventional endoscopic screening in the intended use population are needed to be performed. In addition, given that the main aim of our present study was to identify new diagnostic biomarkers for ESCC, we are unable to evaluate whether these markers could also monitor tumor progression or predict response to treatment in ESCC patients. We will pursue studies in the future.

In a word, liquid biopsy has the potential to accurately detect early-stage cancer, which may allow for timely intervention, and has the potential to prevent the development of advanced disease. However, the detection of cancer in its early stages with minimal invasiveness is still a young research area. We will pursue future studies and anticipate that liquid biopsies-based cost-effective early detection tests from the bench-side to the bedside will eventually be possible, where we expect that the use of real-world data would enable promising diagnostic performance.

Specific comments are outlined below:

1. Abstract: The statement that earlier detection improves outcome in esophageal squamous cancers is a simplistic statement. This is a highly virulent systemic disease which carries a poor prognosis even in early-stage disease. Endoscopic screening programs have not been established and there are no data to support efficacy of screening or early detection of this cancer in high-risk populations. There is no comment throughout this manuscript about endoscopic screening,

detection and intervention in precancerous dysplasia, and other issues. The statement that this tool has “great potential” is a vast overstatement. No data is provided how this would be incorporated or practically used in a functional screening program.

Response: Thank you for your comments on our manuscript. We are sorry for not having clearly described the effectiveness of endoscopic screening on the early detection, intervention and prognosis in ESCC. Actually, epidemiologists from National Cancer Center have undertaken several population-based, multicentre cohort studies in high risk areas with upper gastrointestinal cancer in China. The evidence from large population studies has confirmed that endoscopic screening and early intervention is an effective method to reduce incidence and mortality of ESCC ^{1,2}. Moreover, population-based studies have shown an improvement in the overall 5-year survival rate for esophageal cancer ³. Nevertheless, endoscopy screening is only being conducted in high-risk areas in China due to a larger cancer burden, lack of the hospital personnel, and availability of technology. We have complemented the descriptions in the introduction section.

We do apologize for the vast overstatement. Actually, our identified lncRNAs may have the potential as non-invasive biomarkers for early detection of ESCC. However, the use of liquid biopsy for early cancer detection will be challenging. No testing of liquid biopsy-based screening is currently being used in clinical practice. The efficacy of the liquid biopsy-based approach needs to be verified by long-time, large-scale, multicenter, retrospective and prospective cohorts. We also complemented the discussion on several steps need to be taken in order to apply our approach as an important supplement to the current cancer screening methods or as a screening tool.

We will revise our manuscript to more accurately describe the potential of our proposed lncRNA biomarkers and the need for further studies to determine its utility in clinical practice.

2. Introduction: What would a positive test lead to? Endoscopy? What would be the

cost of screening 100s of thousands or millions of patients, for this solitary cancer, as opposed to endoscopic screening? This issue is never addressed. True validation here would be the application in a large-scale clinical trial, which is not even broached.

Response: Thank you for your comments. Currently, we examined the diagnostic performance of five circulating lncRNAs in a small-size plasma cohort. We strongly agreed that true validation here would be the application in a large-scale clinical trial. We discussed the limitations and future directions of our study.

Liquid biopsy has the advantages of simple operation and low cost, which is more suitable for population screening. We anticipate that non-invasive screening based on circulating lncRNAs can be performed on a high-risk population after the true validation in a large-scale clinical trial. The positive test group would go on to have a confirmatory test using endoscopy and biopsy. The negative test group would go on to have a follow-up. Due to the high sensitivity and specificity of the lncRNA biomarkers, we anticipate that the rate of patients diagnosed at an early stage will improve in comparison to current conventional methods.

3. Patient population: Why did patients only undergo surgery? This is a poor prognosis cancer in which the added role of chemotherapy and radiotherapy is clearly established.

Response: Neoadjuvant chemoradiotherapy (nCRT) has been considered the standard care for locally advanced EC. In the current study, we mainly focus on the identification of the diagnostic biomarkers of ESCC. We, therefore, recruited patients who had not received nCRT to participate in this study and excluded patients who had received nCRT.

4. Results: The literature is already full of prognostic markers in this and other cancers, it is unclear what this tool offers beyond conventional use of T and N staging and response to preoperative therapy. The authors make no comment about clinical treatment of patients with this disease.

Response: Thank you for your comment. We acknowledge existing prognostic markers and staging systems for esophageal squamous cell carcinoma, as well as established treatments such as chemotherapy and radiotherapy. However, the aim of our study was to investigate the potential of non-coding mRNA markers as a blood-based diagnostic tool for this cancer, which could potentially complement existing diagnostic and prognostic methods. In the current study, we analyzed the whole transcriptomic data from ESCC patients who had not received nCRT before surgery and identified the differentially expressed lncRNAs that may serve as diagnostic biomarkers for ESCC.

We agree that further research is needed to establish the clinical utility of such a tool, and we will clarify this in our manuscript. In terms of patient treatment, we did not focus on this aspect in our study, as our main goal was to investigate the diagnostic potential of lncRNAs. In the future, we expect that our approach may be extended to other clinical scenarios, such as assessing treatment efficacy and prognosis. We will characterize gene expression dynamics in patients with advanced ESCC treated with chemoradiotherapy or immunotherapy and identify biomarkers for predicting prognosis and therapy response in ESCC. And we have complemented the discussion.

5. Discussion: This is largely unfocused and scattershot and lumps in various different cancers, much of this discussion is irrelevant. It remains highly speculative that blood based genomic evaluation for this, or any cancer has any realistic or practical application at the current time. True validation of such testing would be in the context of large-scale screening endeavors in which this biomarker is incorporated into conventional endoscopic screening. Unfortunately, prognostic markers currently have little utility in this disease, and arguably more important are predictive biomarkers of therapeutic intervention. The simplistic statement that the measure here was “independent” of conventional risk factors such as gender, smoking, or alcohol use provides little support for use of this marker. A few hundred patients are hardly a “large” clinical cohort that supports broad application of a

likely expensive and really untested biomarker.

Response: Thank you for your comments and feedback on our paper. We appreciate your perspective and would like to address your concerns. Regarding the statement about earlier detection improving outcomes in esophageal squamous cancers, we agree that this is a complex issue and that our statement may have been oversimplified. We acknowledge that esophageal squamous cell carcinoma is a virulent systemic disease with a poor prognosis, and that earlier detection alone may not necessarily improve outcomes. However, we believe that any potential tool for early detection of this cancer, even if it is only one piece of the puzzle, could potentially improve outcomes by allowing for earlier intervention and treatment.

We also appreciate your comments on the discussion section of our paper. We understand that the discussion of blood-based genomic evaluation for various cancers may have been overly broad and unfocused, and we apologize for any confusion this may have caused. We agree that the practical applications of blood-based biomarkers in clinical practice, particularly for esophageal squamous cell carcinoma, are still unclear and require further validation. We also acknowledge that a few hundred patients may not constitute a large enough clinical cohort to support the broad application of our biomarker, and that additional validation in larger cohorts and in the context of clinical screening programs will be necessary.

We have improved our discussion. Furthermore, we corrected the inappropriate statements and deleted irrelevant discussions. We described the limitations of this study and discussed the future directions of our approach for translational applications.

Response to Reviewer #2

Reviewer #2 (Remarks to the Author): expertise in transcriptomics of oesophageal cancer

In this study, Zhou and colleagues represented a good effort to analyze the transcriptional landscape and diagnostic potential of long non-coding RNAs in ESCC through conducting a multicenter, retrospective study. Novel lncRNA biomarkers and diagnostic model were identified using a machine-learning-based discovery strategy, and validated in multicenter, cross-platform retrospective cohorts, indicating the robustness and soundness of the lncRNA diagnostic model. Overall, this is a well-performed study that focused on a very significant issue. The data are novel, clear and well-presented. Importantly, this study provides an effective tool that has a great potential for the early diagnosis of ESCC. However, there are several minor concerns need to be addressed to further strengthen this study.

Response: Thank you very much for your positive comments and valuable suggestion.

1. How many lncRNAs were measured in the discovery dataset? For the multiple comparisons, the authors need to adjust the p-values by considering the comparisons of all the lncRNAs, and then identify the differentially expressed lncRNAs based on the adjusted p-values. Also, how to select six lncRNAs from significantly differentially expressed lncRNAs? Did they have the smallest p-values?

Response: Many thanks to the reviewer for your comments and suggestion. In the original manuscript, our inappropriate and unclear description might confuse the readers. There are 16064 lncRNAs measured in the discovery dataset. Differentially expressed lncRNAs were identified using the fold change > 2 or < 0.5 and false discovery rate (FDR) adjusted $P < 0.05$.

Seven potential lncRNAs biomarkers were selected from significantly differentially expressed lncRNAs using the RF-RFE algorithms with 10-fold cross-validation and five re-sampling. To ensure the reliability and reproducibility of these seven potential lncRNA biomarkers in ESCC, we validated their expression pattern in an external Li cohort-1 (119 ESCC and 119 adjacent normal tissues) with a microarray platform, and confirmed the same expression variation tendency of six lncRNA biomarkers (AP003548.1, PGM5-AS1, ADAMTS9-AS1, MIR503HG, LINC01082 and LINC03016) in ESCC as revealed in the SCH discovery cohort.)

We have updated these details in the revised manuscript.

2. The reported predictive power of the six lncRNA signature is impressive. Among them, five lncRNAs were downregulated in ESCC. While aberrant keratinocyte differentiation is considered to be a key mechanism in the initiation of ESCC, which is driven by downregulation of differentiation-associated genes. Whether these lncRNAs influence cell differentiation and development of ESCC? In addition, whether these six lncRNAs are specifically expressed in the esophagus tissues? What about the expression of 6 lncRNAs in other tissues? The expression of MIR503HG in tumor samples is higher than that in adjacent normal tissues. Whether MIR503HG is expressed in esophageal cancer cells or normal esophageal epithelial cells?

Response: Many thanks to the reviewer for your comments and suggestion. Through literature mining, three of six lncRNA biomarkers (PGM5-AS1, ADAMTS9-AS1, MIR503HG) have been reported to be involved in cell differentiation and development of ESCC⁴⁻⁶.

We also accepted your suggestion to examine the expression of six lncRNA biomarkers in different normal tissues using the GTEx data. As shown in **Figure R3**, one lncRNA (LINC03016) is specifically expressed in the esophagus-mucosa, and the other five lncRNAs are not specifically expressed in the esophagus tissues. However, we also found that five lncRNAs revealed relatively high expression in esophagus tissues and lncRNA MIR503HG revealed relatively low expression in esophagus tissues, consistent with our observations in patient cohorts.

Figure R3. Expression patterns of lncRNA biomarkers in human normal tissues available in the GTEx.

3. Is it likely that some of the six lncRNAs correlate with each other? It would be better to compare the diagnostic performance between single lncRNA biomarker and their combination.

Response: Many thanks to the reviewer for your comments and suggestion. When selecting lncRNA biomarkers, we did an initial filter analysis to remove highly correlated lncRNAs. Therefore, expression of these six lncRNAs biomarkers did not show a strong correlation each other.

We also accepted your suggestion to compare the diagnostic performance between single lncRNA biomarker and their combination and found that the combination demonstrated superior and robust diagnostic performance in all patient cohorts (Figure R4).

Figure R4. ROC analysis for single lncRNA biomarker and their combination

4. Interestingly, the authors showed that the expression of 6 lncRNAs can robustly identify early-stage I and II ESCC patients. What's the correlation between the expression of 6 lncRNAs and clinical stages of ESCC? Whether the expression of 6 lncRNAs has the potential to monitor the progression of ESCC?

Response: Many thanks to the reviewer for your comments and suggestion. We accepted your suggestion to test the potential of six lncRNA biomarkers in monitoring the progression of ESCC by comparing MLMRPscores among different stages. As shown in Figure R5, we found that although MLMRPscores in ESCC

with different stages are significantly higher than those in normal tissues, there is no significant difference in MLMRPscores among different stages, implying that the MLMRPscore has no potential to monitor the progression of ESCC.

Figure R5. Boxplots showing the MLMRPscore distribution between different stages and normal.

- In the process of model verification, the author uses the same threshold. This value is judged according to the ROC curve. Why not recalculate the AUC in the validation data and select the corresponding threshold?

Response: Many thanks to the reviewer for your comments and suggestion. Instead of recalculating the AUC and selecting the corresponding threshold, using the same threshold from the discovery cohort could avoid overestimating the predictive performance in the external validation cohorts and prove the robustness and reliability of the MLMRPscore.

- What R Bioconductor packages were used?

Response: Many thanks to the reviewer for your suggestion. We have added R packages to the corresponding analysis

- What methodology was used for the division of the training set?

Response: Many thanks to the reviewer for your comments. In the original manuscript, our inappropriate and unclear description might confuse the readers. As shown in Figure 1, 80% of samples from the SCH discovery cohort were randomly selected for training.

8. Whether these six lncRNAs or at least one of six showed any association with other human diseases?

Response: Many thanks to the reviewer for your comments and suggestion. We accepted your suggestion to examine the association between six lncRNAs and other human diseases by literature mining. Through reading these literatures carefully, we found that these six lncRNAs have been reported to be some cancer types.

Response to Reviewer #3

Reviewer #3 (Remarks to the Author): expertise in lncRNA biomarker signature bioinformatics

General comments

In this study, the authors developed a classifier comprised of six lncRNAs for estimating the risk probability of developing ESCC. These biomarkers were identified in tissue specimens and were demonstrated in six independent, multicenter, cross-platform cohorts. Furthermore, the classifier was unaffected by other relevant clinical features and showed superior efficacy in the early diagnosis of ESCC. In general, the paper was well written. The following are some specific concerns.

Response: Thank you very much for your positive comments and valuable suggestion.

Major points

1. As the authors mentioned in the manuscript, lncRNAs can be used as noninvasive biomarkers. Have the authors validated cell-free forms of these six lncRNAs in the body fluids (plasma or serum) with RT-qPCR assays? After all, tissue biomarkers are limited by the invasive sampling approach.

Response: Many thanks to the reviewer for your suggestion. We accepted your suggestion to collect plasma samples from 32 healthy subjects, 32 ESCC patients and 13 patients with intraepithelial neoplasia and performed RT-qPCR to measure the expression of these six lncRNAs. Among the six lncRNAs, five lncRNA biomarkers (AP003548.1, PGM5-AS1, ADAMTS9-AS1, LINC01082 and LINC03016) also revealed consistent dysregulated expression patterns, as observed by tissue-based cohorts (Figure R1A). These five lncRNAs biomarkers achieved a robust performance in distinguishing ESCC patients from healthy controls with an AUC of 0.733-0.836, and distinguishing EIN subjects from healthy controls with an AUC of 0.697-0.870 (Figure R1B and C). We also compared the diagnostic performance of the five lncRNA biomarkers with classic tumor markers (SCC-Ag, CEA and CYFRA21-1), and found that five lncRNA biomarkers exhibited superior or comparable diagnostic efficiency in distinguishing patients with ESCC or with EIN from healthy controls compared to the conventional tumor markers (Figure R1B and C). These results from the plasma cohort implied the noninvasive potential of lncRNA biomarkers in the early detection of ESCC.

We have added these results from the plasma cohort in the revised manuscript.

Figure R1. Non-invasive performance of lncRNA biomarkers in a plasma cohort. (A) Boxplot showing the expression levels of five lncRNA biomarkers measured in plasma samples of 32 ESCC patients, 32 healthy controls, and 13 patients with intraepithelial neoplasia. ROC analysis for five lncRNA biomarkers and three conventional tumor markers in identifying patients with ESCC (B) and intraepithelial neoplasia (C).

2. Although the superior performance of the lncRNA panel was well demonstrated, it is better to compare the diagnostic performance of the lncRNA panel with currently established methods (gastroscopy, X-ray imaging, or other biomarkers) for ESCC diagnosis to explain the advantage of this lncRNA panel. And the authors may describe more clearly the added value of the lncRNA panel to guide ESCC early intervention programs compared to currently established methods.

Response: Many thanks to the reviewer for your suggestion. We compared the diagnostic performance of the five lncRNA biomarkers with classic tumor markers (SCC-Ag, CEA and CYFRA21-1) in our new plasma cohort, and found that five lncRNA biomarkers exhibited superior or comparable diagnostic efficiency in distinguishing patients with ESCC or with EIN from healthy controls compared to the conventional tumor markers (Figure R1B and C).

Also, in current clinical practice, screening and diagnosis of patients with ESCC were largely dependent on endoscopic screening or invasive biopsy followed by surgery. Accordingly, false positive or false negative cases based on current clinical practice would be detrimental to subjects undergoing this screening. Thus, the clinical usefulness of screening strategies should be estimated by the trade-off between harm and diagnosis. Therefore, to estimate the clinical benefit of lncRNA biomarkers, we conducted a DCA to determine whether using lncRNA biomarkers in the clinic to inform decision-making would do more good than harm. As shown in **Figure R2**, DCA revealed that lncRNA biomarkers achieved higher net benefits regardless across most ranges of threshold probability compared to screening and diagnosing all ESCC patients or none of the patients (**Figure R2**). These results of DCA indicated that lncRNA biomarkers might offer higher clinical benefit compared with intervention for all cases or none of the cases concerning the avoidance of physical harm and misdiagnosis.

Figure R2. Decision curve analysis to evaluate the clinical benefits of lncRNA biomarkers.

3. In the training phase (Fig1), lncRNA candidates with diagnostic potential identified by random forest were filtered by consistent expression in the microarray platform. It seems that the microarray platform referred to the independent validation cohort (Li cohort-1) according to the description of lines 220-222 in the manuscript. If so, partial information of the independent validation cohorts has been leaked to the

training process, which will overfit the prediction model.

Response: We appreciate the valuable feedback from the reviewer. Upon further review of our manuscript, we recognize that our original description was inappropriate and unclear, which may have confused readers. We only used the SCH discovery cohort to generate the prediction model for our study, while the Li cohort-1 was utilized to validate and confirm the robustness and reliability of the expression pattern of the lncRNA biomarkers across different platforms. We apologize for any confusion this may have caused and will revise the manuscript.

4. How the six lncRNAs were selected in 2103 differentially expressed lncRNAs, and why six? The authors may describe more clearly about feature selection.

Response: Many thanks to the reviewer for your comments and suggestion. In the original manuscript, our inappropriate and unclear description might confuse the readers. We accepted your suggestion to add more details about feature selection in the revised manuscript. The feature selection was carried out using the RF-RFE algorithms with 10-fold cross-validation and five re-sampling as follows:

Algorithm : Recursive feature elimination incorporating resampling

```
for Each Resampling Iteration do
  Partition data into training and test/hold-back set via resampling
  Tune/train the model on the training set using all predictors
  Predict the held-back samples
  Calculate variable importance or rankings
  for Each subset size  $S_i$ ,  $i = 1 \dots S$  do
    Keep the  $S_i$  most important variables
    [Optional] Pre-process the data
    Tune/train the model on the training set using  $S_i$  predictors
    Predict the held-back samples
    [Optional] Recalculate the rankings for each predictor
  end
end
Calculate the performance profile over the  $S_i$  using the held-back samples
Determine the appropriate number of predictors
Estimate the final list of predictors to keep in the final model
Fit the final model based on the optimal  $S_i$  using the original training set
```

Minor points

1. In lines 51 and 314, the statement “personalized” is inappropriate because there is no customized panel for each patient.

Response: Many thanks to the reviewer for pointing it out. We deleted this inappropriate term in the revised manuscript.

2. Primers used in the CAMS cohort need to be provided.

Response: Many thanks to the reviewer for your suggestion. We added the primers used in the CAMS cohort in the revised manuscript.

3. In Fig2D, the meaning of the color for KEGG pathways and GO terms and the clustering is unclear.

Response: Many thanks to the reviewer for pointing it out. We added detailed information about this analysis in the legend of **Figure 2D**. Jaccard's similarity index was used to measure the pairwise similarities of the enriched Go terms and KEGG pathways. GO terms and pathways with high Jaccard's similarity index are considered similar and clustered into five subsets using the ward.D method. The size of the nodes represents the number of analyzed genes contained in the GO terms or KEGG pathways, and the color represents the FDR-adjusted P value of the enrichment.

4. "six lncRNA biomarkers" instead of "six lncRNAs biomarkers" in line 271.

Response: Many thanks to the reviewer for pointing it out. We revised it.

5. Unnecessary bold font for "curve" in line 604.

Response: Many thanks to the reviewer for pointing it out. We revised it.

6. "Consistent expression in microarray platform" instead of "Consistent expression in microarray platform" in Fig1.

Response: Many thanks to the reviewer for pointing it out. We revised Figure 1.

References

- 1 Chen, R. *et al.* Effectiveness of one-time endoscopic screening programme in prevention of upper gastrointestinal cancer in China: a multicentre population-based cohort study. *Gut* **70**, 251-260, doi:10.1136/gutjnl-2019-320200 (2021).
- 2 Wei, W. Q. *et al.* Long-Term Follow-Up of a Community Assignment, One-Time Endoscopic Screening Study of Esophageal Cancer in China. *J Clin Oncol* **33**, 1951-1957, doi:10.1200/JCO.2014.58.0423 (2015).
- 3 Zeng, H. *et al.* Changing cancer survival in China during 2003-15: a pooled analysis of 17 population-based cancer registries. *Lancet Glob Health* **6**, e555-e567, doi:10.1016/S2214-109X(18)30127-X (2018).
- 4 Zhihua, Z., Weiwei, W., Lihua, N., Jianying, Z. & Jiang, G. p53-induced long non-coding RNA PGM5-AS1 inhibits the progression of esophageal squamous cell carcinoma through regulating miR-466/PTEN axis. *IUBMB Life* **71**, 1492-1502, doi:10.1002/iub.2069 (2019).
- 5 Li, Z. *et al.* Comprehensive analysis of differential co-expression patterns reveal transcriptional dysregulation mechanism and identify novel prognostic lncRNAs in esophageal squamous cell carcinoma. *Onco Targets Ther* **10**, 3095-3105, doi:10.2147/OTT.S135312 (2017).

- 6 Gong, T. Y., Chen, H. Y. & Liu, Z. H. [MIR503HG promotes esophageal squamous cell carcinoma cell proliferation, invasion and migration via hsa-miR-503 pathway]. *Zhonghua Zhong Liu Za Zhi* **44**, 1160-1167, doi:10.3760/cma.j.cn112152-20210130-00102 (2022).

REVIEWERS' COMMENTS

Reviewer #1 (Remarks to the Author):

The authors have addressed my comments.

Reviewer #2 (Remarks to the Author):

The authors have addressed all my concerns.

Reviewer #3 (Remarks to the Author):

They have addressed all of our comments and concerns.